# Multimodal Strategy in Localized Merkel Cell Carcinoma: Where Are We and Where Are We Heading?

**DOI:** 10.3390/ijms221910629

**Published:** 2021-09-30

**Authors:** Gianluca Ricco, Elisa Andrini, Giambattista Siepe, Cristina Mosconi, Valentina Ambrosini, Claudio Ricci, Riccardo Casadei, Davide Campana, Giuseppe Lamberti

**Affiliations:** 1Department of Experimental Diagnostic and Specialized Medicine (DIMES), Alma Mater Studiorum, University of Bologna, 40138 Bologna, Italy; gianluca.ricco@studio.unibo.it (G.R.); elisa.andrini2@studio.unibo.it (E.A.); valentina.ambrosini@unibo.it (V.A.); giuseppe.lamberti8@unibo.it (G.L.); 2NET Team Bologna—ENETS Center of Excellence, 40138 Bologna, Italy; Cristina.mosconi@aosp.bo.it (C.M.); Claudio.ricci6@unibo.it (C.R.); Riccardo.casadei@unibo.it (R.C.); 3Radiation Oncology, IRCCS Azienda Ospedaliero—Universitaria di Bologna, 40138 Bologna, Italy; giambattista.siepe@aosp.bo.it; 4Department of Radiology, IRCCS Azienda Ospedaliero—Universitaria di Bologna, 40138 Bologna, Italy; 5IRCCS Azienda Ospedaliero—Universitaria di Bologna, 40139 Bologna, Italy; 6Department of Internal Medicine and Surgery (DIMEC), Alma Mater Studiorum, University of Bologna, 40138 Bologna, Italy; 7Division of Medical Oncology, IRCCS Azienda Ospedaliero—Universitaria di Bologna, 40138 Bologna, Italy

**Keywords:** Merkel cell carcinoma, surgery, radiotherapy, chemotherapy, immunotherapy, avelumab, pembrolizumab, adjuvant, neoadjuvant

## Abstract

Merkel cell carcinoma (MCC) is an aggressive neuroendocrine tumor of the skin whose incidence is rising. Multimodal treatment is crucial in the non-metastatic, potentially curable setting. However, the optimal management of patients with non-metastatic MCC is still unclear. In addition, novel insights into tumor biology and newly developed treatments (e.g., immune checkpoint inhibitors) that dramatically improved outcomes in the advanced setting are being investigated in earlier stages with promising results. Nevertheless, the combination of new strategies with consolidated ones needs to be clarified. We reviewed available evidence supporting the current treatment recommendations of localized MCC with a focus on potentially ground-breaking future strategies. Advantages and disadvantages of the different treatment modalities, including surgery, radiotherapy, chemotherapy, and immunotherapy in the non-metastatic setting, are analyzed, as well as those of different treatment modalities (adjuvant as opposed to neoadjuvant). Lastly, we provide an outlook of remarkable ongoing studies and of promising agents and strategies in the treatment of patients with non-metastatic MCC.

## 1. Introduction

Merkel cell carcinoma (MCC) is a rare and aggressive neuroendocrine malignancy of the skin, initially described in 1972 [1]. As a result of the progressively aging population, changes in risk factors exposition (ultraviolet light exposure and Merkel cell polyomavirus [2,3,4,5]) and improvements in diagnostic accuracy, an increasing trend in MCC incidence has been observed [6,7,8,9]. MCC presents as an asymptomatic rapidly expanding violaceous nodule and has a higher mortality rate (approximately 33–46%) compared with other skins cancer, including melanoma [10,11,12]. In patients with localized MCC, 5-year survival and recurrence-free survival (RFS) are 78% and 65%, respectively, in node-negative and 54% and 48%, respectively, in node-positive disease [2,3,4,5,6,7,9,10,13,14,15]. Several prognostic factors have been identified, including tumor size, male sex, immunosuppression, advanced age, tumor primary site (worse for head and neck, better in occult primary), node status, lymphocytes infiltration, expression of p63 (a member of p53 family) in stage I–II, lymphovascular invasion [16,17,18,19].

Surgery plays a crucial role in localized and locoregional stages, but the high recurrence rate and the aggressiveness of MCC require novel integrated therapeutic strategies to improve survival and to reduce the risk of relapse. Alongside with radiotherapy (RT) and chemotherapy, the promising results of immune checkpoint inhibitors (ICIs) in advanced disease have raised interest in shifting these agents in earlier stages MCC, such as either the neoadjuvant or adjuvant strategy. We analyze the evidence supporting the current treatment recommendations of localized MCC with a focus on potentially ground-breaking future strategies.

## 2. Staging

MCC is an aggressive tumor that shows neuroendocrine features (e.g., somatostatin receptor expression) together with a high mitotic index, tumor necrosis and vascular invasion. Staging is of paramount importance in planning a treatment strategy and because the site of primary tumor remains unknown in up to 25% of cases [20]. The initial staging of the disease should include a CT scan and/or ^18^F-fluoro-deoxy-glucose positron emission tomography (^18^F-FDG-PET) [21]. Indeed, ^18^F-FDG-PET showed high sensitivity and specificity in several studies and is the preferred method for staging in patients with MCC [22,23]. The phase II study Merkel PET Protocol (MP3) is currently evaluating efficacy and safety of chemoradiotherapy in ^18^F-FDG-PET stage II and III MCC (NCT01013779). Interestingly, the study showed that staging PET have a moderate to high impact on MCC management in approximately one-third of cases, affecting treatment decisions in 27.6% of cases, mainly due to upstaging that occurred in 25.9% of patients, with no cases of downstaging [24]. These data suggest the crucial role of staging PET in pre-treatment work-up of locoregional MCC.

The presence of neuroendocrine features in MCC is best studied with PET using somatostatin analogs labeled with positron emitters (^68^Ga-labeled dodecane tetraacetic acid (DOTA)-peptides) and may be useful to identify those patients who can benefit from peptide receptor radionuclide therapy (PRRT), although this therapeutic option is under investigation and not yet approved in this patient population [25].

The sentinel lymph node mapping (SLNM) is a staging method, which consists of biopsy of any lymph node that receives direct lymphatic drainage from the primary tumor [26]. It can be especially useful to plan surgery or delineate RT treatment volumes in MCC in anatomically challenging sites (e.g., midline or head and neck primaries).

## 3. Surgery

Surgery is often the first-line treatment for localized MCC, as complete excision significantly improves overall survival (OS) [27]. Moreover, surgery may be used for the diagnosis and initial management of the truly localized MCC. Removal of the primary tumor should be achieved with 1 to 2 cm free margins investing fascia of muscle, or pericranium when clinically feasible [23]. On the other hand, reduced local excision and the presence of residual tumor cells in specimens are risks factors of relapse and death when adjuvant RT (aRT) is not performed [28,29]. Conservative surgery followed by aRT can be proposed in MCC of the head and neck or in frail patients not eligible for appropriate wide local excision [30,31]. Alternative techniques, used to spare surrounding healthy tissues, include Mohs surgery, modified Mohs surgery and excision with complete circumferential peripheral and deep margin assessment [32,33]. Mohs surgery is performed by removing subsequent thin layers of tumor tissue, until cancer cells are not detected histologically in specimens peripheral and deep margins on frozen sections [34]. In the modified Mohs surgery, outer margins, basal and central portions of specimens are evaluated using paraffin-embedded sections [35]. However, irrespective of the type of surgery, sentinel lymph node biopsy (SLNB) should be carried out before definitive local excision, which can alter lymphatic drainage. Moreover, reconstructive techniques, including flaps and skin grafts, should reduce extensive tissue movements in order to perform SLNB appropriately and avoid delay in aRT start [36,37].

Since 35.4% of patients with MCC present with nodal disease and 50 to 70% of patients develop regional lymph node metastases, the approach to nodal disease is crucial for the management of patients with newly diagnosed MCC [27,38]. SLNB is considered a diagnostic and therapeutic tool for all patients with clinically and radiologically negative nodes (cN0) [39]. Given the high rate of false-negative lymph nodes (14 to 30% according to different studies), especially in MCC of the head and neck, the use of immunohistochemistry with anti-cytokeratins antibodies (e.g., anti-CK20) and intraoperative imprint cytology are used to recognize micro-metastases [40,41]. Sentinel lymph node status is strongly associated with recurrence, distant relapse and survival and can thus drive the subsequent treatment strategy [40,42,43,44,45,46]. In patients with positive lymph nodes at SLNB, the best strategy between complete lymph node dissection (CLND), CLND with aRT and aRT alone is still unclear, since complications of CLND and of aRT can overlap and worsen each other, especially in elderly and frail patients. Available evidence suggests no difference in regional control, disease-free survival (DFS) and MCC-specific survival for SLN-positive patients with MCC who received CLND, CLND plus aRT or nodal aRT alone, suggesting that a monotherapy approach could be sufficient in this setting [39,47,48]. In contrast, a recent analysis from the National Cancer Data Base (NCDB) showed a survival benefit in SLN-positive MCC patients who undergo aRT compared to those who did not [49]. Nevertheless, positive non-SLNs (i.e., metastasis detected in nodal specimens after CLND) are associated with worse survival and higher risk for distant metastases and can be useful to identify high-risk patients, which could benefit from adjuvant systemic therapy [47].

In clinically node-positive disease, CLND is the first-line of treatment in most of the cases, after diagnosis confirmation by fine-needle aspiration or core biopsy. aRT is recommended in the case of involvement of more lymph nodes or extracapsular extension [27]. In-transit metastases, defined as skin lesions between the primary tumor and draining regional nodal basin, can be treated with conservative surgery (if limited in number), RT, isolated limb perfusion or infusion, or electrochemotherapy (for patients with head and neck skin metastases that do not fit any of the previous approaches), as alternatives to otherwise aggressive surgical procedures [50,51].

## 4. Chemotherapy

MCC is a chemosensitive tumor, thus agents such as platinum salts plus etoposide and various alkylating agents have been used alone or in combination in the advanced setting [52]. In the perioperative setting, chemotherapy is not associated with improved OS after resection of MCC in patients with locoregional disease. A retrospective study involving 237 patients with locoregional MCC showed no survival improvement with adjuvant chemotherapy in the overall cohort and in the subgroup of 76 patients with nodal involvement [53]. Similarly, an analysis of 6908 cases from the NCDB showed no survival benefit from adjuvant chemotherapy in patients with stage I–III MCC [54]. The phase II TROG 96:07 study evaluated the efficacy of the chemoradiation therapy in patients with localized MCC (confined to the primary site and nodes) at high risk of recurrence (i.e., relapse after initial surgical resection, nodal disease, size of primary >1 cm or gross residual disease after surgery) [55]. Among 53 patients, 38 (72%) received chemoradiation therapy as adjuvant treatment, whereas 15 patients (28%) received definitive treatment. The study failed to demonstrate a survival benefit, and chemoradiation was associated with high rates of grade 3 and 4 adverse events (AEs), such as skin toxicities (63%), neutropenia (60%) and sepsis (40%), thrombocytopenia (10%), and anemia (6%) [56]. In agreement with this, a systematic review of 34 studies suggested that adjuvant chemoradiation (*n* = 301) was not superior to aRT alone (*n* = 1689) in terms of local control, recurrence and survival but increased acute and chronic toxicities [57].

MCC arising from the head and neck region are characterized by worse prognosis given the irregular nodal drainage of this site that makes it difficult to achieve an optimal postoperative staging and management [58,59]. An analysis of the NCDB data of 4815 patients with MCC of the head and neck showed that adjuvant chemoradiotherapy and aRT were associated with similarly improved survival compared to surgery alone [60]. Nevertheless, adjuvant chemoradiotherapy was associated with improved OS compared to aRT alone in patients with high-risk features such as positive margins after surgery (HR: 0.48, 95% CI:0.25–0.93; *p* = 0.03), tumor size ≥ 3 cm (HR: 0.52, 95% CI: 0.30–0.90; *p* = 0.02) and male sex (HR: 0.69, 95% CI: 0.50–0.94; *p* = 0.02).

Given the unproven benefit on survival and associated toxicities, the addition of adjuvant chemotherapy to aRT is not routinely recommended, but its use can be considered on a case-by-case basis [61,62].

## 5. Radiotherapy

Currently, aRT to both the primary site and the nodal basin is widely recommended by guidelines for the management of both node-positive and node-negative localized MCC as it is associated with improved survival [23]. However, the role of aRT after complete resection of an MCC is still debated as improvement has been observed only in local control in some studies, while in others, it was also associated with prolongation of survival, e.g., in patients with high-risk MCC [63,64,65,66,67,68,69,70,71]. These results can be explained by selection bias and unmeasured confounders, as most of the evidence is derived from registry studies, such as the NCDB or the Surveillance, Epidemiology, and End Result (SEER) registry, or small retrospective series. However, different considerations about aRT must be made depending on the presence or not of nodal metastases.

### 5.1. aRT after Resection of Node-Negative MCC (Stage I–II)

A prospective randomized trial showed that prophylactic regional RT after resection and aRt on the tumor bed of stage I MCC improved regional recurrence rates compared to observation alone (0% versus 16.7%, respectively; *p* = 0.007), but not OS or RFS, possibly due to premature trial interruption after the introduction of SLNB in routine clinical practice [72].

An analysis of 6908 cases from the NCDB demonstrated that aRT was associated with a significant benefit in OS in stage I (hazard ratio (HR) = 0.71; *p* < 0.001) and stage II MCC (HR: 0.77; *p* < 0.001) [54], as also confirmed by a recent metanalysis [73]. However, high-risk factors for local and nodal recurrence should be considered, since patients with low-risk small lesions have a reduced risk for local recurrence as compared to those with high-risk factors or positive lymph nodes [73,74,75]. aRT is associated with improved survival in patients with head and neck MCC (in whom also local control is improved [76]) and with stage I–II MCC, in both immunocompromised and immunocompetent patients [77]. Thus, aRT alone may also be indicated in SLN-negative patients in the presence of risk factors such as immunosuppression and head and neck sites, while clinical observation or dosing antibodies against MCPyV should be reserved for patients with tumor size less than 1 cm, widely excised, and without risk factors (e.g., immunosuppression, lymph vascular invasion, microscopic or grossly positive margins and head and neck site) [23]. Finally, many Australian institutions consider it acceptable in clinical practice to deliver definitive RT with the aim of improving local control in a select group of locally advanced patients with MCC in the presence of immunosuppression or severe comorbidities or if extensive surgery would cause disfigurement or delay of a potential adjuvant treatment [11,78,79,80].

### 5.2. aRT after Resection of Node-Positive MCC (Stage III)

Several studies have shown the role of aRT in node-positive MCC. The previously mentioned analysis from the NCDB showed no OS benefit associated with aRT in patients with stage III MCC [54]. On the other hand, another study showed an improvement in 3-year disease-specific survival (DSS) among node-positive patients (76.2 vs. 48.1%; *p* = 0.035), as well as an improvement of locoregional control in patients with either pathologically or clinically positive nodes who received aRT compared to those who did not [70]. Andruska and colleagues showed that aRT on regional lymph nodes, including both definitive RT and RT after CLND, improved regional RFS (HR: 0.07; *p* = 0.003), distant RFS (HR: 0.28; *p* = 0.01), DFS (HR: 0.23; *p* = 0.002) and DSS (HR: 0.23; *p* = 0.03) without any difference between definitive RT and RT after CLND [81]. In addition, RT after CLND improved DFS and DSS among patients with high-risk factors such as extra-nodal involvement, ≥2 positive lymph nodes and foci of nodal disease >0.5 cm. In the case of clinically negative LNs but positive SLNB, NCCN guidelines recommend aRT on the nodal basin after CLND if multiple nodal metastases, high tumor burden and extracapsular extension are present [23]. On the contrary, in the case of clinically positive lymph nodes, aRT on the draining nodal basin is recommended after CLND if poor prognostic factors are detected, such as multiple nodal involvement and/or extracapsular extension, but not in the case of low tumor burden on SLN. In this setting, participation in a clinical trial should be proposed if available [23].

## 6. Immunotherapy

The immune system has a pivotal role in MCC development and in defining the clinical outcome of patients, as suggested by the fact that MCC is more common in immunocompromised patients, MCPyV is integrated into tumor DNA in 80% of cases, and UV DNA damage typical of MCC results in a high tumor mutational burden (defined as the number of non-synonymous mutations per megabase) [14,82,83,84]. Furthermore, similar to what observed in melanoma, head and neck squamous cell carcinoma and non-small cell lung cancer [85,86,87,88], higher PD-L1 expression on MCC cells is positively associated with MCPyV DNA integration, with the presence of tumor-infiltrating lymphocytes (TILs), and improved survival [89,90]. Recently, ICIs have become the new standard first-line treatment for patients with metastatic MCC, as they have been shown to improve OS compared to platinum-etoposide chemotherapy [91,92,93,94,95], but only avelumab, a fully human monoclonal antibody directed against the programmed death ligand 1 (PD-L1), received FDA and EMA approval, following the results of the JAVELIN Merkel 200 trial.

Considering the exciting results in the advanced setting, the role of immunotherapy has been explored and is currently being explored in earlier stages as well. The randomized, multicenter DeCOG/AGO study (NCT02196961) of ipilimumab, an human monoclonal antibody against the cytotoxic T lymphocyte-associated antigen-4 (CTLA-4), versus observation after complete resection of primary or locoregional metastatic MCC enrolled 40 patients [96]. The primary endpoint, DFS was not significantly different between the two arms (HR: 1.8; *p* = 0.48), but a 4-fold higher AEs rate was observed in the treatment arm compared to the observation arm, leading to premature discontinuation of the study. Other ICIs targeting the interaction between the programmed death 1 (PD-1) and PD-L1, such as pembrolizumab (a humanized monoclonal antibody against PD-1) and avelumab, respectively, are being investigated in the adjuvant setting in ongoing studies (NCT03712605, NCT03271372, and NCT04291885).

Neoadjuvant therapy has the advantage of controlling micrometastatic disease before surgery and to potentially induce downstaging, allowing patients to undergo a less invasive surgery, which is particularly crucial in MCC originating from the head and neck region, as shown by some case reports [97]. The shortcoming of this approach is the potential development of progression of disease in non-responders before surgery. Taking this into account, the neoadjuvant approach could be recommended in those patients in which the surgical procedure bears a high risk of morbidity (e.g., borderline-resectable disease, involvement of neurovascular structures or multiple lymph node basins) or those at high risk of systemic recurrence of MCC (e.g., resectable stage IV disease, multiple in-transit metastases, multiple large and/or rapidly growing nodes) [98]. In this setting, the Checkmate 358 trial evaluated the safety and efficacy of nivolumab in high-risk resectable MCC (stage IIA-IV) [99]. Nivolumab 240 mg was administered intravenously on day 1 and 15, whereas surgery was performed on day 29. Among 39 patients, 3 did not undergo surgery for tumor progression or AEs, but 47.6% achieved a pathologic complete response (pCR). Among tumor specimens from 26 patients evaluated by central pathologic review, 12 (46.2%) showed a pCR, and 4 patients (15.4%) a major pathologic response (MPR), defined as a residual viable tumor ≤10%, resulting in a combined pCR+MPR rate of 61.5%. Responses were observed regardless of viral status and PD-L1 expression. RFS at 12 months was 100.0% in patients with pCR versus 59.6% in patients without pCR, and RFS at 24 months was 88.9% versus 52.2%, respectively. Median RFS was not reached. A total of 18 out of 39 patients (46.2%) experienced any-grade AE (with grade 3-4 AEs reported in three patients), the most common being skin reactions (10.3%). The available evidence is summarized in Table 1.

## 7. On the Horizon: Future Perspectives

Given the rising incidence and the aggressive behavior of MCC, efforts are being made to improve the chance of cure in the non-metastatic setting and most are focused on the inclusion of ICIs or other immune-oncology approaches. Ongoing clinical trials are summarized in Table 2.

With respect to ICIs in the adjuvant setting, the ADMEC-O phase II, randomized, open label trial (NCT02196961) is currently evaluating the efficacy of adjuvant nivolumab monotherapy in completely resected MCC patients without metastases versus observation. In the I-MAT trial, a randomized, placebo-controlled, phase II study (NCT04291885), adjuvant avelumab is being investigated in patients with stage I–III MCC following local and regional treatment, including surgery and/or RT. Furthermore, another randomized, placebo-controlled, phase III trial (NCT03271372) of adjuvant avelumab in stage III patients with MCC who have undergone surgery and/or RT is currently ongoing. The phase III trial STAMP study (NCT03712605) is evaluating efficacy of adjuvant pembrolizumab in completely resected patients with stage I–III MCC also in combination with RT. Another randomized, phase I, pilot study (NCT03798639) is currently evaluating safety and activity of nivolumab in combination with aRT compared to nivolumab and ipilimumab in MCC stage III patients, as RT may serve as a priming agent for immune system recognition of residual tumor cells [100,101,102,103]. Since a proportion of MCC may express somatostatin receptors, the phase Ib/II GoTHAM trial (NCT04261855) is currently evaluating efficacy and safety of either 177Lu-DOTA-octreotate (LuTate) or conventional fractionated radiotherapy both in combination with avelumab in patients with metastatic MCC.

In the neoadjuvant setting, immunotherapy left many open issues that should be addressed by ongoing and future studies such as the optimal treatment duration, a better stratification of patients who could benefit from this strategy and the potential need of a postoperative therapy (aRT, additional immunotherapy or targeted therapy) in patients with poor pathologic or clinical response. The combination of Lenvatinib (a multikinase inhibitor with immunomodulatory effects), plus pembrolizumab is being evaluated in several trials [104] given the potential synergistic antitumor activity, as shown in different tumor types [105,106,107]. A phase II single arm clinical trial (NCT04869137) is currently enrolling patients to receive neoadjuvant therapy with Lenvatinib and pembrolizumab before surgical resection planned within 2–4 weeks and followed by single-agent pembrolizumab with or without RT.

Other remarkable immunotherapy strategies under investigation in MCC include toll-like receptor (TLR) agonists and genetically modified viruses. TLR agonists promote inflammatory changes in the tumor microenvironment, increasing the infiltration of CD8+, CD4+ T cells, antigen-presenting cells and the expression of chemokines and cytokines-related genes, in order to reverse immunosuppressive mechanisms and facilitate local and systemic immune responses [108]. These agents may promote tumor regression before surgery and/or RT. Intratumoral G100, a TLR4 agonist, demonstrated promising results in terms of responses and tumor regression in a pilot study in patients with locoregional and metastatic MCC [109]. Talimogene laherparepvec (TVEC) is a genetically modified herpes simplex virus, which is administered intralesionally and induces expression of the granulocyte-macrophage colony stimulating factor in order to potentiate tumor-antigen presentation by dendritic cells [110]. TVEC is under investigation in patients with localized or locally advanced MCC before surgery and/or RT, as single-agent or in combination with ICIs in order to determine the response rate and immune changes in the tumor microenvironment (NCT02819843, NCT02978625, NCT03458117) [111,112].

Therapeutic vaccines targeting tumor-associated antigens or viral oncoproteins, such as ST and LT in MCPyV-driven MCC patients are also being developed as these could be particularly appealing in the perioperative setting of skin cancers, such as MCC, to reduce tumor volume, eradicate potential micrometastases, and finally help prevent recurrence [113]. Clinical trials investigating vaccines in advanced or locoregional MCC are currently ongoing (NCT04160065, NCT04246671). Interestingly, IFx-Hu2.0 is an experimental intralesional vaccine, which induces the expression of the emm55 gene by tumor cells and may lead to a broad immune response that can potentially overcome primary and secondary resistance to ICIs in patients with advanced skin cancer, including MCC (NCT04853602).

## 8. Conclusions

The optimization of currently available treatment modalities (RT, immunotherapy, targeted therapy) and their combination, as well as the evaluation of newly developed options, is essential to improve the survival and quality of life of patients with early-stage MCC, as these patients are often frail and comorbid but may be cured. Improving knowledge about tumor biology is crucial to set up and integrate new treatment strategies in the management of patients with MCC. As this is a rare disease, patients with MCC should be directed to high-volume referral centers to best manage this challenging disease, within a multidisciplinary tumor board, and offer access to clinical trials whenever available.

## Figures and Tables

**Table 1 ijms-22-10629-t001:** Summary of available data in the management of non-metastatic Merkel cell carcinoma.

Authors and Reference	Type of study/Treatment	*n*	Stages	OS	DSS	DFS	LR-RFS	D-RFS	RFS
Andruska et al. [22]	Retrospective; WLE margins	79	I–II		HR = 0.16 (0.04–0.61); *p* = 0.007	HR = 0.42 (0.21–0.86); *p* = 0.02	(regional): HR = 0.28 (0.11–0.75); *p* = 0.01	HR = 0.30 (0.08–0.99); *p* = 0.049	
Perez et al. [40]	Retrospective; LND; RT; LND + RT	71	III	at 3 years 71% (23–92); 67% (46–81); 69% (40–86); *p* = 0.72	at 3 years 100%; 93% (73–98); 93% (59–99); *p* = 0.63		regional, at 2 years 100%; 93% (77–98); 94% (67–99); *p* = 0.60	at 2 years 86% (33–98); 78% (58–89); 82% (55–94); *p* = 0.68	
Lee et al. [41]	Retrospective; LND vs. RT	179	III		HR = 0.4 (0.1–1.3); *p* = 0.1	HR = 0.7 (0.3–1.6); *p* = 0.4	regional HR = 1.6 (0.3–7.5); *p* = 0.6	HR = 0.9 (0.2–3.5); *p* = 0.9	
Fang et al. [42]	Retrosective; RT vs. LND ± RT	24	III	(at 2 years) 63% vs. 59%	(at 2 years) 53% vs. 79%; *p* = 0.9		regional, at 2 years 78% vs. 73%; *p* = 0.8		
Cramer et al. [43]	Retrospective	447	III	OBS vs. LND + RT: HR = 3.54 (1.36–9.18); *p* < 0.05 LND vs. LND + RT: HR = 2.54 (1.03–6.27); *p* < 0.05					
Bhatia et al. [48]	Retrospective; S vs. S + CT	6908	I–II–III	HR = 0.79 (0.60–1.05; *p* = 0.11) (stage I) HR = 1.14 (0.89–1.45; *p* = 0.30) (stage II) HR = 0.97 (0.85–1.12; *p* = 0.71) (stage III)					
Poulsen et al. [49]	Prospective; S + CRT	53	I–II	3-years OS: 76% (63.5–88.7)			75% (62.7–87.5)	76% (63.4–88.7)	65% (50.8–78.5)
Poulsen et al. [50]	Retrospective; S vs. S + CT	102	I–II	HR = 0.64 (0.34–1.2; *p* = 0.16)	HR = 0.68 (0.33–1.43; *p* = 0.31)				
Hasan et al. [51]	Systemic review; S + CRT vs. S + RT	4475	I–II–III	89% vs. 90% (1 year OS); 73% vs. 70% (3 years OS)			80% vs. 84% (1 year LC); 69% vs. 65% (3 years LC)		
Chen et al. [54]	Retrospective; S + CRT vs. S	4815	I–II–III	HR = 0.62 (0.47–0.81)					
Jabbour et al. [58]	Retrospective; S + RT vs. S	82	I–II	HR = 0.39 (0.18–0.82); *p* = 0.013		S: HR = 0.39 (0.2–0.75); *p* = 0.004			
Kim et al. [59]	Retrospective; S vs. S + RT:	747	I–II–III	HR = 1.28 (1.01–1.63); *p* = 0.046	HR = 0.94 (0.68–1.31); *p* = 0.72				
Mojica et al. [60]	Retrospective; S + RT vs. S	1187	I–II–III	HR = 0.85 (0.75–0.96); *p* = 0.0122					
Han et al. [62]	Retrospective; S + RT vs. S	40	I–II	HR = 0.24 (0.07–0.85); *p* = 0.027					
Lewis et al. [63]	Meta-analysis; S + RT vs. S	1254	I–II–III	HR = 0.42 (0.14–1.24); *p* = 0.11	HR = 0.73 (0.26–2.08); *p* = 0.56		local: HR = 0.33 (0.18–0.58); *p* < 0.001 regional: HR = 0.19 (0.09–0.38); *p* < 0.001		
Strom et al. [64]	Retrospective; S + RT vs. S	171	I–II–III	HR = 0.53 (0.31–0.93); *p* = 0.03	HR = 0.42 (0.26–0.70); *p* = 0.001	HR = 0.42 (0.26–0.7); *p* = 0.001	local: HR = 0.18 (0.07–0.46); *p* < 0.001 locoregional: HR = 0.28 (0.14–0.56); *p* < 0.001		
Servy et al. [65]	Retrospective; S + RT vs. S	87	I	HR = 0.16 (0.05–0.51); *p* = 0.0019	HR = 0.19 (0.07–0.5); *p* < 0.001				
Jouary et al. [66]	Multicentric prospective randomized; S + RT vs. S	83	I	HR = 1.23 (0.39–4.54); *p* = 0.98					
Petrelli et al. [67]	Meta-analysis; RT vs. no RT	17,179	I–II	HR = 0.81 (0.75–0.86); *p* < 0.001		HR = 0.45 (0.32–0.62); *p* < 0.001	local: HR = 0.3 (0.22–0.42) locoregional: HR = 0.21 (0.14–0.33)	HR = 0.79 (0.49–-1.14)	
Fields et al. [69]	Retrospective; S + RT vs. S	364	I–II–III				local: HR = 0.53 (0.13–2.19); locoregional: HR = 0.48 (0.24–0.97)		
Takagishi et al. [70]	Retrospective; S + RT vs. S	46	I	HR = 0.9 (0.23–3.5)			local: HR = 0.15 (0.03–0.76)		
Andruska et al. [74]	Retrospective; rLNRT	72	III		HR = 0.23 (0.06–0.9); *p* = 0.03	HR = 0.23 (0.09–0.58); *p* = 0.002	regional: HR = 0.07 (0.01–0.4); *p* = 0.003	HR = 0.28 (0.11–0.76); *p* = 0.01	
The Checkmate-3581	Clinical trial S + Nivolumab vs. S	39	IIA-IV						12 months-RFS: 100.0% vs. 59.6%, 24 months-RFS: 88.9% vs. 52.2% (HR = 0.12 [0.01–0.93])

**Table 2 ijms-22-10629-t002:** Ongoing clinical studies of interest enrolling patients with non-metastatic Merkel cell carcinoma (source: clinicaltrials.gov; last accessed: 30 July 2021).

Identifier Name	*n*	Phase	Arm/Arms	Primary Outcome Measure	Estimated Primary Completion Date
NCT04869137 Neoadjuvant Lenvatinib Plus Pembrolizumab in Resectable Merkel Cell Carcinoma	16	2	Experimental: Lenvatinib plus Pembrolizumab	Pathological Complete Response	May, 2024
NCT02196961 Prospective Randomized Trial of an Adjuvant Therapy of Completely Resected Merkel Cell Carcinoma (MCC) With Immune Checkpoint Blocking Antibodies Every 3 Weeks for 12 Weeks Versus Observation	180	2	Experimental: Nivolumab; Control: observation	Disease-free survival (DFS) at 12 months	August, 2022
NCT04291885 A Randomised, Placebo-controlled, Phase II Trial of Adjuvant Avelumab in Patients With Stage I–III Merkel Cell Carcinoma	132	2	Experimental: Avelumab; Control: placebo	Recurrence-free survival (RFS)	December, 2025
NCT03271372 A Multicenter, Randomized, Double-Blinded, Placebo-Controlled, Phase 3 Trial of Adjuvant Avelumab (Anti-PDL-1 Antibody) in Merkel Cell Carcinoma Patients With Lymph Node Metastases	100	3	Experimental: Avelumab; Control: placebo	Relapse-free survival (RFS)	August, 2024
NCT03712605 STAMP: Surgically Treated Adjuvant Merkel Cell Carcinoma With Pembrolizumab, a Phase III Trial	500	3	Experimental: Pembrolizumab; Pembrolizumab plus radiotherapy Control: observation; radiotherapy	Overall survival (OS) and recurrence free survival (RFS)	October, 2023
NCT03798639 Randomized, Multi-Institutional Pilot Study of Nivolumab and Radiation Therapy Versus Nivolumab and Ipilimumab as Adjuvant Therapy for Merkel Cell Carcinoma	7	1	Experimental: Nivolumab plus radiotherapy; Control: Nivolumab plus ipilimumab	Percentage of patients completing 12 months of treatment	December, 2021
NCT04160065 Phase 1 Trial of Intralesional Immunotherapy With IFx-Hu2.0 Vaccine in Patients With Advanced Merkel Cell Carcinoma or Cutaneous Squamous Cell Carcinoma	20	1	IFx-Hu2.0	Number of Grade 3-5, Treatment-Related Adverse Events	December, 2021
NCT04246671 Phase 1/2 Expansion Cohorts Trial of Intravenous Administration of TAEK-VAC-HerBy Vaccine Alone and in Combination With HER2- and PD-1/PD-L1 Antibodies in Patients With Advanced HER2-expressing Cancer	45	1/2	Experimental: TAEK-VAC-HerBy Stage 1 (1 × 107 Inf.U; 1 × 108 Inf.U; 1 × 109 Inf.U; 1 × 1010 Inf.U) TAEK-VAC-HerBy Stage 2 (Trastuzumab + TVH; T-DM1 + TVH; Trastuzumab + Pertuzumab + TVH; Her2 + TVH + PD1/PD-L1)	Patients with Dose Limiting Toxicity (DLT)	December, 2022
NCT02643303 A Phase 1/2 Study of In Situ Vaccination With Tremelimumab and IV Durvalumab (MEDI4736) Plus the Toll-like Receptor Agonist PolyICLC in Subjects With Advanced, Measurable, Biopsy-accessible Cancers	102	1/2	Experimental: IV Durvalumab + IT/IM polyICLC (Phase I, Cohort 1A); IV Durvalumab + IV Tremelimumab + IT/IM polyICLC(Phase I, Cohort 1B); IV Durvalumab + IT Tremelimumab + IT/IM polyICLC(Phase I, Cohort 1C) Phase 2 Cohort: Durvalumab + Tremelimumab + polyICLC.	Progression-Free Survival (PFS) at 24 weeks	December, 2021
NCT04853602 IFx-Hu2.0 Expanded Access Program		Expanded Access	IFx-Hu2.0		
NCT03435640 A Phase 1/2, Open-label, Multicenter, Dose Escalation and Dose Expansion Study of NKTR-262 in Combination With Bempegaldesleukin (NKTR-214) and in Combination With Bempegaldesleukin Plus Nivolumab in Patients With Locally Advanced or Metastatic Solid Tumor Malignancies	64	1/2	Experimental: Doublet: NKTR-262 + bempegaldesleukin; Triplet: NKTR-262 + bempegaldesleukin + Nivolumab	Safety, Tolerability and Efficacy	September, 2021
NCT02819843 A Phase II Randomized Trial of Intralesional Talimogene Laherparepvec (TALIMOGENE LAHERPAREPVEC) With or Without Radiotherapy for Cutaneous Melanoma, Merkel Cell Carcinoma, or Other Solid Tumors	19	2	Experimental: Talimogene laherparepvec plus RT Talimogene laherparepvec without RT	Overall subject level response	June, 2021
NCT02978625 A Phase II Study of Talimogene Laherparepvec Followed by Talimogene Laherparepvec + Nivolumab in Refractory T Cell and NK Cell Lymphomas, Cutaneous Squamous Cell Carcinoma, Merkel Cell Carcinoma, and Other Rare Skin Tumors	68	2	Experimental: Talimogene laherparepvec plus nivolumab	Response rate to talimogene laherparepvec aloneBest overall response rate to combination therapy	June, 2021
NCT03458117 A Phase I, Open Label, Single Arm, Single Centre Study to Evaluate Mechanism of Action of Talimogene Laherparepvec (T-VEC) in Locally Advanced Non-melanoma Skin Cancer	20	1	Experimental: Talimogene Laherparepvec (T-VEC)	Change from Baseline local immune effects	February, 2022

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
