# Peer review of "Multimodal Strategy in Localized Merkel Cell Carcinoma: Where Are We and Where Are We Heading?"

_ijms, 2021, doi:10.3390/ijms221910629_

Round 1

Reviewer 1 Report

Dear Authors

well done, a nice review of MCC and its multi-modal management

I have several comments which I think will make it a better and more comprehensive review

In the introduction and in the 1st sentence of the "surgery" section you say

"surgery is the cornerstone in localised and locoregional stages" .  I think this is a (subconscious) pro-surgical bias.  I agree that it is often the 1st line of management (if even for management) and certainly think your statement is more true for  localized but NOT so true for loco-regional disease.

In the same vein, surgery may be used for diagnosis and /or initial management of the truly localized primary is how i think the surgical section should start.  As surgeons are (rightly or wrongly) the gate-keepers they will always have first bite of the cherry (even when patient should have been discussed in MDT).

In the Chemotherapy section 

you grossly mis-represent the TROG 9607 trial where you sell it as an adjuvant CRT trial.  This is NOT the case - in 15 of the 53 patients in this trial, the CRT was DEFINITIVE.  Furthermore, this trial went on to inform the MP-3 trial (which you don't reference at all).  This was a better tolerated form of definitive (and adjuvant) CRT and also contained a nested trial about the value of FDG-PET staging in this disease - a valuable addition to your manuscript.  in MP-3 50% of patients had definitive CRT.

Radiation Section

please change Currently aRT "on" both  to "to both".  Second sentence should read "prolongation of survival".  The last sentence in 4.1 should start with the word "Finally" not lastly.  there is an important reference you have missed in addition to 73 + 74.  Please add Sundaresan P Clin Onc 2012

Immuno section reads well

In future perspectives - you must mention DOTA scans for both imaging and therapy. About 60% MCC DOTA avid and may be targeted with DOTA labelled radiation.  This is being investigated in the randomised Australian GOTHAM trial

I strongly feel your  review would benefit from a short Staging section about FDG and DOTA PET imaging for optimal staging.  Also about use of sentinel node mapping to better plan RT fields or localise darining node for close surveillance or inclusion in RT field (Naehrig D et al JMIRO)

Author Response

See attached file.

Responses to reviewer are in red.

Reviewer 2 Report

Check the misspelling , expecially in the Introduction and Abstract

Author Response

We thank the reviewer for these observation and suggestion. 

We corrected typos.

Reviewer 3 Report

This is an interesting and well written review concerning We reviewed  the current treatment recommendations of localized Merkel cell carcinoma with a focus on potentially ground-breaking future strategies. The review is well structured and conclusions are consistent with the data of the literature.

Author Response

We thank the reviewer for this comment.